# Nondestructive Classification of Soybean Seed Varieties by Hyperspectral Imaging and Ensemble Machine Learning Algorithms

**DOI:** 10.3390/s20236980

**Published:** 2020-12-07

**Authors:** Yanlin Wei, Xiaofeng Li, Xin Pan, Lei Li

**Affiliations:** 1Northeast Institute of Geography and Agroecology, Chinese Academy of Sciences, Changchun 130102, China; weiyl18@mails.jlu.edu.cn (Y.W.); lilei@iga.ac.cn (L.L.); 2College of Electronic Science and Engineering, Jilin University, Changchun 130012, China; 3Changchun Institute of Technology, Changchun 130012, China; panxin@iga.ac.cn

**Keywords:** hyperspectral imaging, correlation coefficient matrix, ensemble machine learning algorithms, random subspace linear discriminant

## Abstract

During the processing and planting of soybeans, it is greatly significant that a reliable, rapid, and accurate technique is used to detect soybean varieties. Traditional chemical analysis methods of soybean variety sampling (e.g., mass spectrometry and high-performance liquid chromatography) are destructive and time-consuming. In this paper, a robust and accurate method for nondestructive soybean classification is developed through hyperspectral imaging and ensemble machine learning algorithms. Image acquisition, preprocessing, and feature selection are used to obtain different types of soybean hyperspectral features. Based on these features, one of ensemble classifiers-random subspace linear discriminant (RSLD) algorithm is used to classify soybean seeds. Compared with the linear discrimination (LD) and linear support vector machine (LSVM) methods, the results show that the RSLD algorithm in this paper is more stable and reliable. In classifying soybeans in 10, 15, 20, and 25 categories, the RSLD method achieves the highest classification accuracy. When 155 features are used to classify 15 types of soybeans, the classification accuracy of the RSLD method reaches 99.2%, while the classification accuracies of the LD and LSVM methods are only 98.6% and 69.7%, respectively. Therefore, the ensemble classification algorithm RSLD can maintain high classification accuracy when different types and different classification features are used.

## 1. Introduction

Soybeans are the world’s leading oil crop and are also used as high-protein foods and feeds. Soybeans and soy foods have attracted widespread attention due to their health benefits [1]. Different soybean seed varieties have different nutrient components (oil, protein, fat, etc.) [2]. Special varieties of soybeans need to be identified and chosen in soybean processing enterprises. On the other hand, the biological characteristics of different soybean seed varieties are also very different. The identification of soybean seed varieties is a critical factor that improves the quality of produced soybean. Therefore, there is a need for a reliable, rapid, and accurate technique to detect soybean varieties.

Traditional seed cultivar identification methods include seedling morphological identification, field planting identification, and electrophoretic analysis [3,4,5]. However, all of these methods have shortcomings, such as a long identification time, a strong dependence on personnel, seed destruction, and popularization difficulties [6]. Because of the advantage of rapid measurements with minimal nondestructive sample preparation, in recent years, hyperspectral imaging technology has been used for classifying soybean varieties [7]. For example, Kong et al. [8] used hyperspectral imaging to classify four varieties of rice seeds, and most of the discriminant models using full spectra and selected wavebands obtained good classification results (over 80%); Liu et al. [9] used hyperspectral imaging to classify soybeans, maize and rice, the spectral data were extracted and then optimal wavelengths were selected.

A hyperspectral image obtained by a hyperspectral imaging system consists of several hundred spectra with high spectral resolution. These bands often have a high spectral redundancy, i.e., the correlation between adjacent bands is high. These redundant bands greatly increase the number of subspace features, not only increasing the computing complexity but also reducing the classification accuracy [10]. Spectral features are often extracted from selected optimal wavelengths as the primary information to the discriminant models in the seed classification. At present, the selection of the optimized band is basically independent of the classification algorithm, which is implemented by principal component analysis (PCA), successive projections algorithm (SPA), etc. [10] or by using a genetic algorithm combined with a classification algorithm [11]. The typical problem with these attribute selection algorithms is that some “specific features” in the sample set may be emphasized or exaggerated in future classification models through screening or combination. Additionally, these “specific features” may be due to the specific batches of seed sample selection and collection. These features cannot be as the main basis for seed classification.

In addition, most of the existing classification methods are obtained by a single classifier, which can easily produce a discriminant deviation [12]. Due to the hundreds of bands contained in a hyperspectral image, a single classifier easily converges during the training process when the number of samples is relatively small, and this convergence may be caused by a specific feature of the sample set being identified. The trained single classifiers have a lower accuracy when used to classify real samples (i.e., a lower generalization ability) [13]. On the other hand, aiming at the specific problem of the hyperspectral classification of soybean seed varieties, the within-class differences of spectral features are relatively small, so a single classifier without attribute filtering during training is more prone to overfitting, which leads to the low actual application value of the developed classification model. For those application fields with many attributes and models that require high generalization capabilities, an ensemble classifier is a good solution [14].

In general, multiclassifier or ensemble-based techniques are preferable to single classifiers because they reduce the possibility of improper selection [15]. The ensemble classifiers combine a set of classifiers that might produce superior classification performance compared to each individual classifier. The ensemble classifiers transform the original classification task into a training process of multiple subclassifiers by dividing in the feature or sample space. Then, multiple subclassifiers are used to reduce overfitting and improve the generalization capabilities. Therefore, it is necessary to introduce ensemble classifier technology for the hyperspectral classification of soybean seed varieties.

In this paper, the random subspace linear discriminant (RSLD) algorithm of ensemble machine learning is used to realize the selection of the band. The weak classifiers obtained by training in the subspace are integrated into strong classifiers. The experimental results show that the RSLD algorithm has better robustness and classification accuracy than a single classifier in the classification of soybean seed varieties.

## 2. Materials and Methods

### 2.1. Hyperspectral Imaging and Data Acquisition

A hyperspectral imaging system covering the 400–1000 nm spectral range was used to acquire images of soybean seeds in reflectance mode (Figure 1). The system consisted of a charge-coupled device (CCD) camera, an imaging spectrograph (Pika XC2, Resonon Inc., Bozeman, MT, USA), a zoom lens (XENOPLAN, F/1.4 FL23 mm, Schneider-KREUZNACH, Bad Kreuznach, Germany), an illumination unit (OSRAM, Munich, Germany) with a 4-lamp lighting system (35 W per lamp, for a total of 140 W of input power and a total radiated power of approximately 5–7 W) and a computer equipped with data acquisition and control software (SpectrononPro software, Resonon Inc., USA). The imaging spectrograph had a spectral range of 400–1000 nm and a 50  μm slit, a spectral resolution of 1.3 nm, and a spatial resolution of 0.15 mm/pixel.

Hyperspectral images of 25 soybean varieties approximately 50 seeds for each variety with 462 spectral bands were collected using a hyperspectral imaging system, as shown in Figure 1, in an optical darkroom (with absorbed material on the wall of the laboratory). First, an optical whiteboard was used for the calibration, and then the soybean seeds were placed in a small plate wrapped with light absorbing material. Hyperspectral images are scanned as the linear translation stage moves.

### 2.2. Image Processing and Spectrum Extraction

Because soybean seeds are spherical, the hyperspectral image of the central area of the sphere will exhibit a “hot spot” effect, as shown in Figure 2I. The edge of the soybean seed kernel will also exhibit a darker slight reflection. The “hot spot” and the dark area at the edge part can be removed by the Otsu threshold method [16] and the morphological erosion operation. The specific processing steps are as follows: (1) the single-band image of a soybean is binarized using the Otsu threshold method to obtain an image, as shown in Figure 2II. In this study, the reflectance image of soybean seeds at 640.08 nm was selected because the seed shows the sharpest outline in the image; (2) the edge of the soybean seeds is extracted using the open operation and the threshold method is used to extract the reflectance “hot spot” at the center of the soybean, as shown in Figure 2III; (3) the image of Figure 2III is subtracted from the Figure 2II image to obtain soybean images without a “hot spot” and with the edges removed, as shown in Figure 2IV; (4) Using the white region in Figure 2IV as the template of the region of interest (ROI), the hyperspectral image of Figure 2V is masked, and the hyperspectral curve of 462 bands is extracted. The obtained hyperspectral data of every pixel in the ROI are corrected with the whiteboard and black reference map using the following formula:(1)$$S_{c}=\frac{S_{0}-S_{b}}{S_{w}-S_{b}}$$
where $S_{0}$ and $S_{c}$ are the raw and corrected spectral images of the sample, respectively; $S_{b}$ is the dark reference image obtained by completely blocking the lens of the hyperspectral imaging camera with an opaque cap; $S_{w}$ is the white reference image acquired by the 99% diffuse reflectance white standard. (5) The hyperspectral reflectance of individual soybean seeds is obtained by calculating the average reflectance of all pixels in each soybean, as shown in Figure 2VI.

### 2.3. Random Subspace Linear Discriminant (RSLD) Algorithm

A classification process based on the hyperspectral features obtained from hyperspectral images is used to classify the soybean seed varieties. In the current study, one ensemble classifier, the random subspace linear discriminant (RSLD) model, is trained and used to distinguish the different soybean seed varieties.

The general algorithm flowchart is shown in Figure 3. Before the RSLD algorithm classification, to improve the efficiency of model training and avoid overfitting, the correlation coefficient matrix (CCM) method of hyperspectral data [10] is used to construct some new band subsets as the initial subspace to reduce the data redundancy.

Next, the workflow of implementing the RSLD algorithm is represented by the part in the dotted box in Figure 3. First, the RSLD algorithm randomizes the learning algorithm by selecting a subset of features (chosen subspace), which are represented as Subspacei (i = 1…k) in Figure 3. The number of band features in each subspace, i.e., the number of predictors, directly affects the classification accuracy of the algorithm and needs to be properly selected. Then, the corresponding linear discriminant (LD) classifiers are trained on the basis of each subspace Subspacei, i.e., LDi (i = 1… k) in Figure 3.

The LD classifier is a supervised learning classification algorithm [17] that projects high-dimensional pattern samples to the best discriminant vector space to extract classification information and compress the feature space dimension. The projection can ensure that the samples have the largest interclass distance and the smallest intraclass distance in the new subspace, which means that the pattern samples have the best separability. Therefore, it is an effective method for feature extraction and classification.

In the classification by the RSLD method, the number of LD learners (k) is another important parameter; it is also the number of combined weak learners that may produce the strong classifier. The number of predictors and LD learners are determined through experimental analysis so that the RSLD model can classify soybean seed varieties more accurately.

Finally, the outputs of models are combined by a majority vote to generate classification results [18]. In other words, RSLD ensemble classifiers integrate random subspace division and the LD analysis scheme to determine a specific discriminant subspace of low dimensions [13,19,20], which is more suitable for hyperspectral image processing with redundant band features. Therefore, the subspace ensemble through the LD method with the majority voting rule is used in the current work.

### 2.4. Model Training and Testing

A training set of models is constructed by randomly selecting spectral data according to a certain proportion (2/3 in this study) from all hyperspectral data of soybean varieties, and the remainder is used as a validation set. In the model training, a repeated 10-fold cross validation strategy is used to obtain the training accuracy [21]. The proper model parameters (e.g., the number of features and classifiers) can be determined by observing the accuracy changes with parameter tuning.

For a hyperspectral sample set $S=\left\{\left(x_{1},y_{1}),(x_{2},y_{2}),\cdots ,(x_{n},y_{n}\right)\right\}$, where n is the number of all samples and $x_{i}$ and $y_{i}$ are the spectral features of soybean seeds and the true class label, respectively, in the process of model training and testing, the following indices are used to characterize the model accuracy:(2)$$E=\frac{1}{n}\sum_{i=1}^{n}\Delta (f(x_{i})\neq y_{i}),$$
(3)$$A=\frac{1}{n}\sum_{i=1}^{n}\Delta (f(x_{i})=y_{i})=1-E$$
where *E* and *A* represent the error rate and accuracy of the classifiers, respectively; Δ is the number of samples that meet the condition in the bracket; f represents the classifier.

## 3. Results and Discussion

### 3.1. Soybean Varieties and Hyperspectral Data

Hyperspectral images of 25 soybean varieties with approximately 50 seeds for each variety were obtained by a hyperspectral imaging system, as shown in Figure 1. Using image preprocessing by the method in Section 2.2, the spectra of each soybean seed were corrected (total 1250 spectral curves). The mean spectra of each variety were calculated and are shown in Figure 4. It can be seen from the figure that the spectral reflectance of all kinds of soybeans is almost less than 0.6. There are obvious absorption valleys between 600 and 700 nm, and there exist some differences in the spectral reflectance of the various varieties after 700 nm.

### 3.2. Determination of the Initial Subspace

The soybean seed images collected by the hyperspectral imaging system used in this paper consist of 462 bands (400–1000 nm) and may have a high spectral redundancy between bands, i.e., high correlation between adjacent bands. In Figure 5, the CCM of a hyperspectral dataset for soybean variety (JY204) is shown. Figure 5a shows the correlation between the 462 bands from a wavelength of 392 nm for JY204 seeds. The brighter areas in the image represent high correlation, and the darker areas represent low correlation. As shown in Figure 5b, the adjacent bands of the data exhibit high correlation, and the average correlation coefficients within diagonal blocks are greater than 0.8 (0.87, 0.92, 0.96, and 0.97). The hyperspectral correlation matrix of other varieties of soybean seeds also has a spatial distribution pattern similar to that of Figure 5.

According to the CCM pattern and the average correlation coefficients within the diagonal blocks, as shown in Figure 5b, the band intervals are divided into four blocks (bands No. 1–90, 91–190, 191–225 and 226–462) before the subspace model training. In each band interval, the primary subspace is constructed by selecting the number of subspace bands at regular intervals according to 20, 30, 40, and 50 of the band numbers, and then the models using these different initial subspaces with the different band numbers are trained by the random subspace method.

### 3.3. Best Model Parameters

The most important of the model parameters required by the random subspace classifier is the number of predictors, i.e., the number of hyperspectral features randomly selected for model training to construct the individual subspace; the second is the number of learners, i.e., the number of classifiers in the ensemble, which has a large impact on the classification accuracy and computational efficiency of the model. The over-trained classification may occur with too large ensembles, and larger ensembles have longer training times for the prediction. The proper number of combined weak learners can achieve better learning and prediction results while considering the efficiency.

Figure 6 shows the variation curve of the algorithm classification mean error (10-fold cross validation with 10 repetitions) with the number of predictors and the number of learners. According to the CCM shown in Figure 5, 10, 20, 30, 40, and 50 features are selected at equal intervals in each matrix block. Since the matrix block in band 191–225 has only 35 features, when 40 and 50 features are selected for each matrix block, 35 features in the matrix block are selected totally, so that in the four types of feature selection, 40, 80, 120, 155, and 185 band characteristics participated in the experiment.

As seen from Figure 6, when the number of predictors is greater than 30 and the number of learners is greater than 25, the cross-validated error tends to stabilize and gives good predictions, reaching below 0.05 and 0.033, respectively. Therefore, there seems to be no advantage in an ensemble with more than 30 predictors or 25 learners; additionally, a larger number of predictors and learners will also consume more computing resources. In the following classification recognition task of this article using the RSLD algorithm, the following parameters were set: the number of predictors: 30 and the number of learners: 25.

### 3.4. Soybean Seed Variety Classification and Validation

In this paper, the RSLD method is used to identify and classify soybean varieties. At the same time, previous studies in the literature have shown that the linear support vector machine (LSVM) and LD methods also have high classification accuracy [22,23]. Linear Discriminants (LD) is a statistical method of dimensionality reduction that provides the highest possible discrimination among various classes [24]. Linear support vector machine (LSVM) is a supervised learning model with associated learning algorithms that analyze data used for classification and regression analysis [25]. We randomly constructed a training set and a validation set from the samples and tested the accuracy of the three algorithms (LSVM, LD and RSLD) after training. Three methods were compared to distinguish the different kinds of soybean seeds (10, 15, 20, and 25 varieties) at the selected different numbers of spectral bands (40, 80, 120, 155, 185, and 462 bands). The classification accuracy is shown in Figure 7.

To further explore and compare the classification performance of the three algorithms in different situations, Table 1 gives the test accuracy and average accuracy for each of the three algorithms in 10 varieties of soybean seeds with different classification characteristics. In all 198 different categories of the data comparison, the accuracy of the RSLD algorithm is slightly less than that of the LD algorithm in only five cases (see the black italic numbers in Table 1), and the final average accuracy is higher than that of the LSVM and LD algorithms in all cases.

Table 2 and Figure 7 also give the average accuracy statistical data and histograms of the three algorithms when 10, 15, 20, and 25 types of soybeans are classified. From Table 2 and Figure 7, it is easy to see that the RSLD algorithm still performs well, except that the individual classification accuracy is slightly lower than the LD model when there are fewer features (40, 80 features), while the others are higher than the LD and LSVM algorithms. It can be seen from the data in the table that if the classification accuracy is required not to be less than 95%, then 155 band features can be selected, i.e., 30 features are selected in each matrix block shown in Figure 5.

In addition, the average accuracy of the three algorithms in the four kinds of varieties and five features (40, 80, 120, 155, and 462) is also given in the last three rows and the last column of Table 2, respectively. For a more intuitive accuracy comparison of the three algorithms in both cases, Figure 8 shows the results of the accuracy polyline. From Figure 8, we can observe that the accuracy of LSVM classification is always the lowest. The classification accuracy of the single classifier LD algorithm gradually increased and reached its highest value when the number of features is 185 but decreased when the number of features increase to 462. This result shows that although the multispectral features can provide more information, too many features can cause overfitting, and the classification accuracy decreases. In contrast, the accuracy of the RSLD classifier is similar or even slightly inferior to a single classifier in the case of fewer features, which provide less information, but as the number of features increases, the classification accuracy gradually improves. After a single classifier reaches its highest point, the RSLD method can still improve the classification accuracy.

In summary, the following conclusions can be drawn: (1) the classification accuracy of the RSLD method is higher than that of the LSVM and LD algorithms in most cases (see Table 1 and Table 2); (2) the classification accuracy of the RSLD algorithm is stable with a change in the characteristics in the same category number of soybean classification and has highest accuracy when features are larger than 40. The accuracy of the LD algorithm is higher than that of LSVM, both of which fluctuate with feature changes (see Figure 7). (3) When all features (462 bands) are used for classification, the RSLD remains stable and slightly improved, but the LD and LSVM methods have the lowest accuracy (see Figure 8), indicating that the characteristics of the selection method in this study have the effect on improving the accuracy of all three methods, especially for the LD. Therefore, for the RSLD, the proper selection of the number of features can reduce the amount of computation while maintaining classification accuracy.

## 4. Conclusions

In this paper, the nondestructive hyperspectral imaging of soybean seeds is used to collect hyperspectral images of different varieties of soybeans. The hyperspectral characteristics of the soybeans are obtained through image preprocessing methods, and the correlation matrix blocks of the spectral characteristics are obtained through correlation analysis. Based on these features, LSVM, LD, and RSLD classification models were obtained using randomly selected training sets. The classification accuracy of each algorithm was obtained through a validation data set. The results show that the RSLD algorithm is superior to the other two algorithms in terms of the stability and accuracy. The RSLD algorithm also has a great advantage when more varieties of soybeans are classified. When using 155 features to classify 15 types of soybeans, the classification accuracy of RSLD reaches 99.2%, while the classification accuracies of LSVM and LD are only 69.7 and 98.6%, respectively. Therefore, the ensemble classification algorithm RSLD can maintain high classification accuracy when different types and different classification features are used. The RSLD method has the potential to perform the high-precision online classification of soybean varieties with the data processing flow proposed in this study.

The RSLD algorithm may have a certain degree of uncertainty when fewer features are provided. In this case, the accuracy of the three algorithms is not suitable for accurately identifying soybean seed categories. To achieve better recognition results, more spectral features are needed. Therefore, in addition to the spectral features themselves, other spectral combination indices and soybean morphological features can also be added as classification attributes. Thus, the RSLD algorithm can achieve better results when the features are richer.

## Figures and Tables

**Figure 1 sensors-20-06980-f001:**
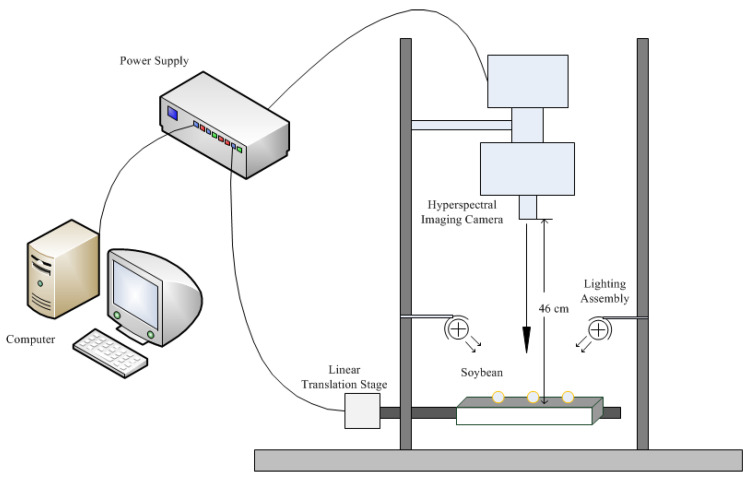
Hyperspectral imaging system.

**Figure 2 sensors-20-06980-f002:**
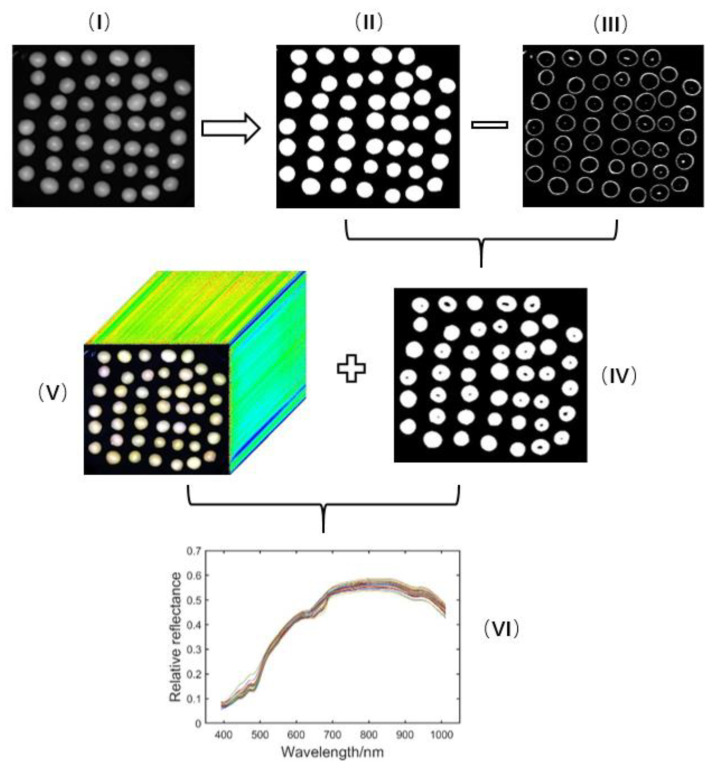
Image processing and spectrum extraction. (**I**) The reflectance image of soybean seeds at 640.08 nm; (**II**) The binarized image by Otsu threshold method; (**III**) The edge of the soybeans and the reflectance “hot spot” extraction by the open operation and the threshold method; (**IV**) The region of interest of the soybean seeds; (**V**) The hyperspectral image of the soybean seeds; (**VI**) The hyperspectral reflectance of individual soybean seeds.

**Figure 3 sensors-20-06980-f003:**
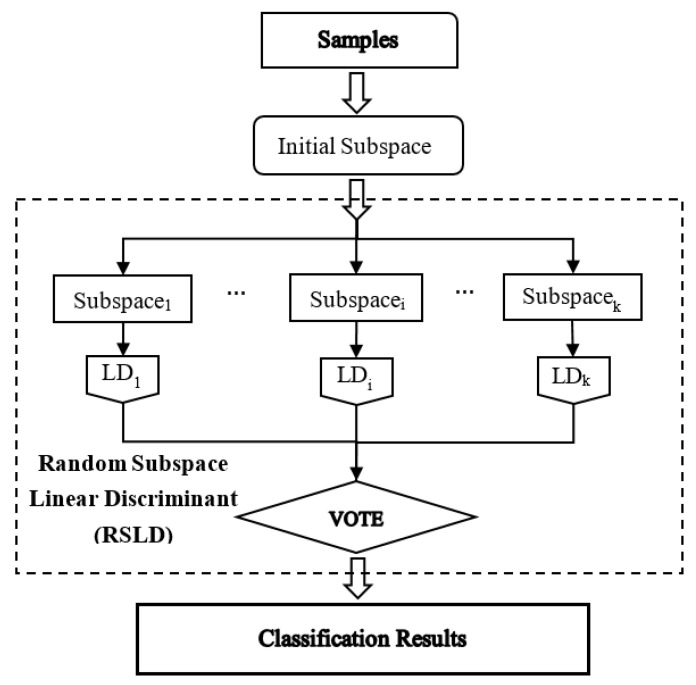
Flowchart of the classification by the random subspace linear discriminant (RSLD) ensemble classifier.

**Figure 4 sensors-20-06980-f004:**
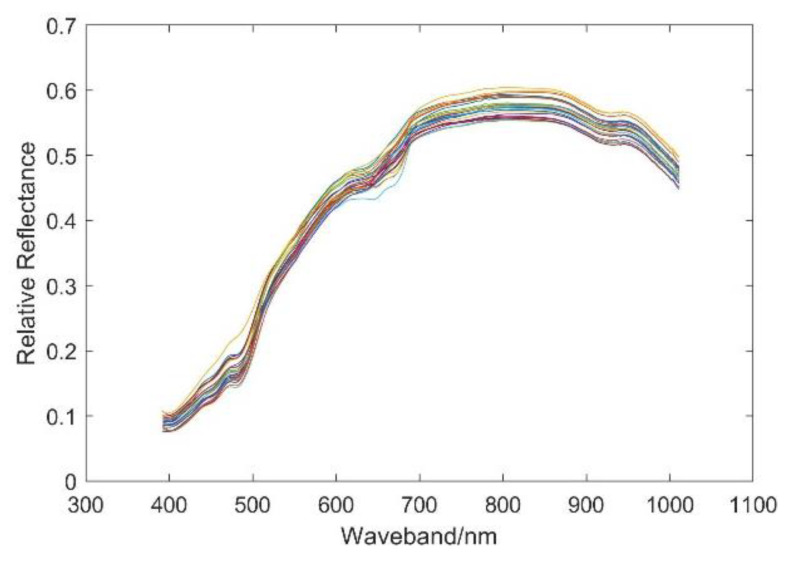
The mean spectrum graph for each of the 25 soybean seed varieties.

**Figure 5 sensors-20-06980-f005:**
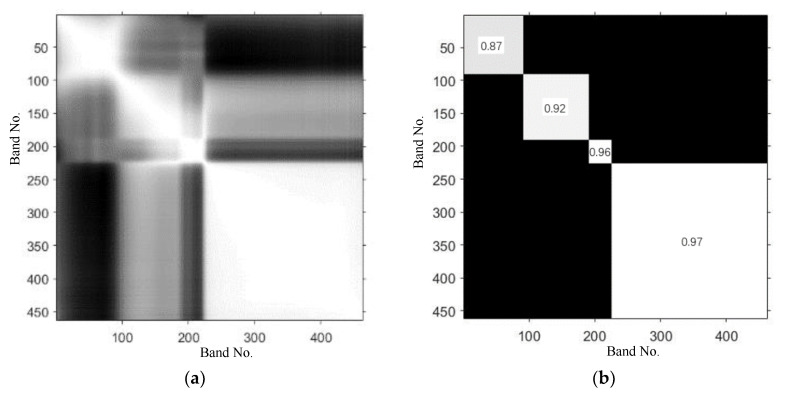
(**a**) The correlation matrix of 462 hyperspectral bands for JY204 soybean seed; (**b**) Average correlations within diagonal blocks.

**Figure 6 sensors-20-06980-f006:**
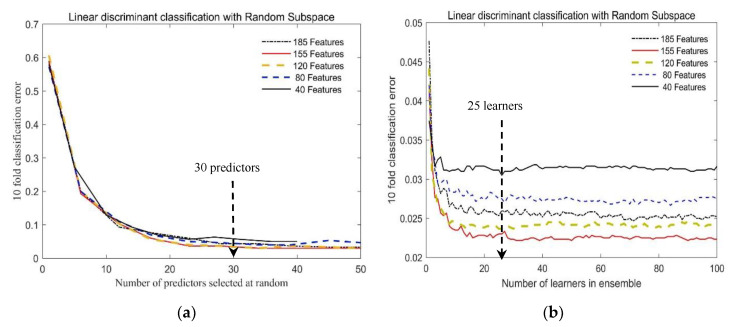
Ten-fold classification error of the RSLD algorithm varies with (**a**) the number of predictors and (**b**) the number of learners in the ensemble using 30 predictors, when 40, 80, 120, 155 and 185 band features are selected to classify 25 soybean varieties according to the correlation matrix.

**Figure 7 sensors-20-06980-f007:**
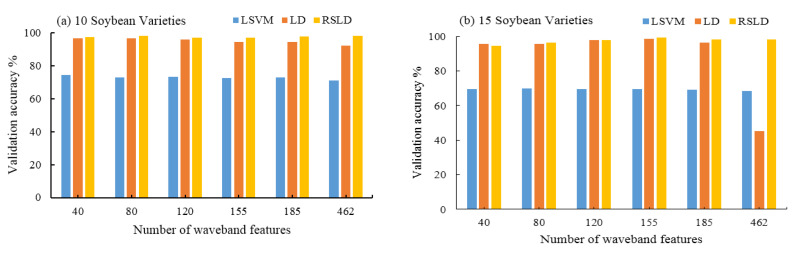
The validation accuracies of 10 (**a**), 15 (**b**), 20 (**c**), and 25 (**d**) soybean varieties classified by the LSVM, LD, and RSLD algorithms using the selected 40, 80, 120, 155, and 185 band features and all spectral band features (462) according to the correlation matrix.

**Figure 8 sensors-20-06980-f008:**
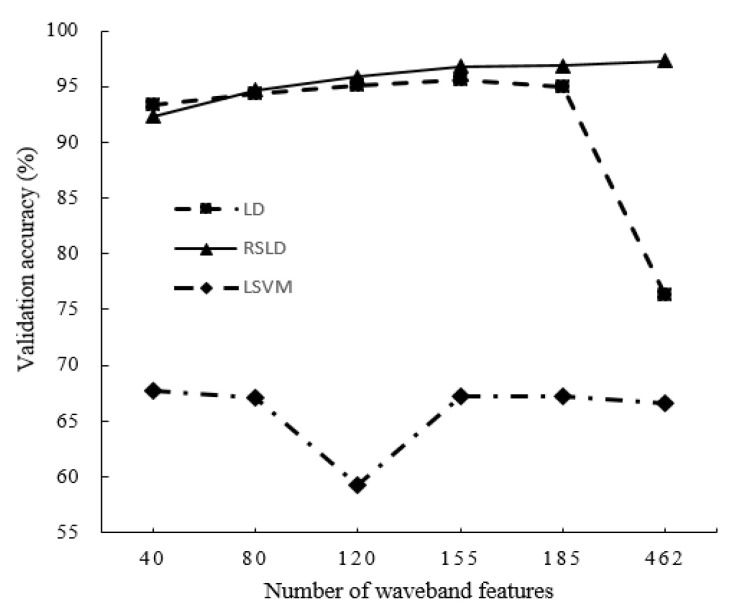
The average classification accuracies by the LSVM, LD and RSLD algorithms versus with the number of band features.

**Table 1 sensors-20-06980-t001:** Classification accuracy (%) of soybean seeds from 10 varieties of seeds by using linear support vector machines (LSVM), linear discriminant (LD) and random subspace linear discriminant (RSLD) algorithms (Algor.) based on all features (462 bands) and the selected 40, 80, 120, 155, and 185 band features (Feat.) according to the correlation matrix.

	Variety	S1	S2	S3	S4	S5	S6	S7	S8	S9	S10	Ave
Feat. & Algor.												
**40 Feat.**	**LSVM**	80.0	78.3	73.3	94.7	58.6	75.8	89.5	66.7	62.5	64.7	**74.4**
	**LD**	100.0	100.0	86.7	100.0	100.0	93.9	100.0	95.2	95.8	94.1	**96.6**
	**RSLD**	100.0	100.0	93.3	100.0	100.0	***90.9***	100.0	95.2	100.0	94.1	**97.4**
**80 Feat.**	**LSVM**	80.0	73.9	66.7	94.7	58.6	75.8	86.8	66.7	62.5	64.7	**73.0**
	**LD**	100.0	100.0	86.7	100.0	96.6	97.0	97.4	95.2	100.0	94.1	**96.7**
	**RSLD**	100.0	100.0	93.3	100.0	100.0	97.0	100.0	95.2	100.0	94.1	**98.0**
**120 Feat.**	**LSVM**	80.0	73.9	73.3	94.7	55.2	75.8	86.8	66.7	62.5	64.7	**73.4**
	**LD**	100.0	100.0	93.3	100.0	96.6	90.9	100.0	95.2	95.8	88.2	**96.0**
	**RSLD**	100.0	100.0	***86.7***	100.0	100.0	93.9	100.0	95.2	100.0	94.1	**97.0**
**155 Feat.**	**LSVM**	80.0	73.9	66.7	94.7	55.2	75.8	86.8	66.7	62.5	64.7	**72.7**
	**LD**	85.0	100.0	93.3	94.7	96.6	90.9	100.0	95.2	100.0	88.2	**94.4**
	**RSLD**	100.0	100.0	***86.7***	100.0	100.0	93.9	100.0	95.2	100.0	94.1	**97.0**
**185 Feat.**	**LSVM**	80.0	73.9	73.3	94.7	51.7	75.8	86.8	66.7	62.5	64.7	**73.0**
	**LD**	95.0	100.0	86.7	100.0	96.6	90.9	92.1	100.0	87.5	94.1	**94.3**
	**RSLD**	100.0	100.0	93.3	100.0	100.0	93.9	100.0	***95.2***	100.0	94.1	**97.7**
**462 Feat.**	**LSVM**	85.0	69.6	66.7	94.7	51.7	75.8	86.8	57.1	62.5	58.8	**70.9**
	**LD**	95.0	100.0	86.7	100.0	96.6	90.9	92.1	100.0	87.5	94.1	**94.3**
	**RSLD**	100.0	100.0	93.3	100.0	100.0	97.0	100.0	***95.2***	100.0	94.1	**98.0**

**Table 2 sensors-20-06980-t002:** Classification accuracy (%) of 10, 15, 20, and 25 soybean seed varieties (Varie.) by using linear support vector machines (LSVM), linear discriminant (LD), and random subspace linear discriminant (RSLD) algorithms (Algor.) based on all features (462 bands) and the selected 40, 80, 120, 155, and 185 band features (Feat.) according to the correlation matrix.

	Feat.	40 Feat.	80 Feat.	120 Feat.	155 Feat.	185 Feat.	462 Feat.
Varie. & Algor.							
**10 Varie.**	**LSVM**	74.4	73.0	73.4	72.7	73.0	70.9
	**LD**	96.6	96.7	96.0	94.4	94.3	94.3
	**RSLD**	97.4	98.0	97.0	97.0	97.7	98.0
**15 Varie.**	**LSVM**	69.7	70.0	69.4	69.7	69.2	68.3
	**LD**	95.6	95.6	97.8	98.6	96.4	45.3
	**RSLD**	94.4	96.4	97.8	99.2	98.1	98.3
**20 Varie.**	**LSVM**	64.3	63.9	63.9	64.3	64.3	65.2
	**LD**	91.0	93.0	92.5	95.0	94.3	76.4
	**RSLD**	88.8	92.1	94.1	95.6	96.1	96.1
**25 Varieties**	**LSVM**	62.6	61.5	30.1	62.1	62.1	61.7
	**LD**	90.3	92.0	93.9	94.5	94.9	89.1
	**RSLD**	88.7	92.3	94.5	95.4	95.6	96.8
**Varie. Average**	**LSVM**	67.8	67.1	59.2	67.2	67.2	66.5
	**LD**	**93.4**	94.3	95.1	95.6	95.0	76.3
	**RSLD**	92.3	**94.7**	**95.9**	**96.8**	**96.9**	**97.3**

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
