# Peer review of "Nondestructive Classification of Soybean Seed Varieties by Hyperspectral Imaging and Ensemble Machine Learning Algorithms"

_sensors, 2020, doi:10.3390/s20236980_

Round 1
Reviewer 1 Report
Dear assistant editor and authors
The paper entitled “Nondestructive Classification of Soybean Seed Varieties by Hyperspectral Imaging and Ensemble Machine Learning Algorithms” will certainly contribute to the precision agriculture research area. This paper presents a novel approach to nondestructive soybean classification based on hyperspectral imaging and ensemble machine learning algorithms. The algorithm random subspace linear discriminant (RSLD) is used to classify soybean seeds, and its performance is compared with two other learners: linear discrimination (LD) and linear support vector machine (SVM). The authors demonstrated that classification algorithm RSLD can maintain high classification accuracy even when different types and different classification features are used. Abstract, Introduction, and Conclusion sections are very informative. Congratulations. But, the authors are requested to clarify certain questions in the methodological part. After this revision, I consider the manuscript appropriated to be published in the Sensors Journal.
I include several additional questions, comments, and suggestions:
- Do images of soybean seeds were collected with the hyperspectral system in the reflectance mode directly or in radiance value, which was converted in reflectance after that?
- Please, inform in Figure 1 what was the height from the hyperspectral system to the stage where seeds were positioned.
- It is not clear in the methodological section how big was the dataset used during hyperspectral data collection. For example, “number of seeds”, “number of seeds varieties”, etc. The authors are requested to provide all these pieces of information in this section. Even though it is informed that hyperspectral images of 25 soybean varieties with approximately 50 seeds for each variety were used, this information appears only in the result section.
- Was the random subspace linear discriminant (RSLD) ensemble classifier developed by the authors, and if so, please inform the language and platform used, or it is implemented in some software?
- The authors said that a training dataset was constructed by randomly selecting spectral data according to a certain proportion (e.g., 2/3). In my opinion, this information is uncertain. Do training and validation set samples proportions variate or fix in 2/3? Does the dataset used for the validation process was not be seen before by the algorithm?
- It was adopted a 10-fold validation with only one repetition? I suggest that the authors consult this paper to verify the possibility to apply several repetitions with 10-fold each one https://www.sciencedirect.com/science/article/abs/pii/S0168169920319591?via%3Dihub
- Please, provide in the methodological section which machine learning metrics for classification tasks were adopted to evaluate the models' performance.
- To adequately compare the performance of the three algorithms (LSVM, LD, and RSLD) in the different experimental conditions (i.e. soybean seeds varieties and numbers of spectral bands selected), I recommend the inclusion of other robust classification metrics, such as Global Accuracy, Precision, Recall, and F1-measure. Additionally, I recommend the construction of ROC (Receiver Operator Curve) - AUC (Area Under The Curve) curve to compare the performance of these algorithms. Please, allow me to suggest this informal reading:
https://towardsdatascience.com/understanding-auc-roc-curve-68b2303cc9c5#:~:text=AUC%20%2D%20ROC%20curve%20is%20a,degree%20or%20measure%20of%20separability.&text=The%20ROC%20curve%20is%20plotted,is%20on%20the%20x%2Daxis.
- I’ve noted that only 17 references were used in the whole manuscript. Please, consider amplifying this number, as this theme, i.e. remote sensing data applied to precision agriculture problems presents a vast related literature.
Kind regards.
Reviewer 2 Report
This paper presents an extremely interesting topic with a new approach. However, a comparison of RSLD with LD and LSVM algorithms is presented, and it is important to explain both LD and LSVM in brief so that the advantages that the application of RSLD algorithm offers can be discussed.
L119. for what stands the "(5)"
Tables 1 and 2 - it needs to be indicated how the accuracy is presented (is it % or ?)
Reviewer 3 Report
The manuscript demonstrates a new method to perform classification of soybean seeds using hyperspectral imaging. In general, the manuscript was well written and easy to follow.
Changes:
1) I disagree with the statement "the classification accuracy of the RSLD method is higher than that of the LSVM and LD algorithms in all cases". This is the only reason I marked "can be improved" for "are the conclusions supported by the results", where the key issue is the wording "in all cases". For example, in Figure 7c for fewer wavebands, LD outperforms RSLD.
Recommendations:
1) Figure 6 appears to indicate the effect of two hyperparameters, # of learners and # of predictors, on the classification error. It appears in each panel, only one of the hyperparameters was varied. Please specify the value of the constant hyperparameter, either in Figure 6 or its caption.
2) A couple of the results appear to be outliers (e.g. Figure 7b LD 462, Figure 7d LSVM 120). I would be interested in seeing some commentary on why those outliers occur.
3) I would recommend displaying validation accuracy in Figure 7 from 0% to 100%, not 40%. Displaying the full range provides a more intuitive comparison of relative performance.
4) I had to read through the paragraph beginning "From Table 1, Table 2, Fig. 7 and Fig 8" more than once. I believe the content is fine other than noted above, but I would reword this paragraph to improve clarity.
Questions
1) I accept the authors' claims about the motivation/need to classify soybean seeds. However, since I am not an expert on soybeans, I have trouble seeing why this is the case. I assume that the variety must be based upon the parent plant, not what the resultant seed would produce. Wouldn't farmers know what they have planted, and wouldn't that simplify classification? Is the concern that farmers might report the wrong (more expensive) variety?
